# Changes in Soil Properties, Microbial Quantity and Enzyme Activities in Four *Castanopsis hystrix* Forest Types in Subtropical China

**DOI:** 10.3390/plants12132411

**Published:** 2023-06-22

**Authors:** Renjie Wang, Jianwei Ma, Huizi Liang, Yubao Zhang, Jisheng Yang, Fengfan Chen, Yong Wang, Wende Yan

**Affiliations:** 1Guangxi Key Laboratory of Superior Timber Trees Resource Cultivation, Nanning 530002, China; wangrj1991@foxmail.com (R.W.); mjw675554649@foxmail.com (J.M.); 2Guangxi Forestry Research Institute, Nanning 530002, China; lianghuizi12345@foxmail.com (H.L.); yjs1769693441@foxmail.com (J.Y.); chenfengfan0402@foxmail.com (F.C.); 3College of Life Science and Technology, Central South University of Forestry and Technology, Changsha 410004, China; yubaozhangzyb@foxmail.com

**Keywords:** plantation, mixed forest, soil physicochemical properties, microorganism quantity, enzyme activities, *Castanopsis hystrix*

## Abstract

It is well established that forest type can have a profound impact on soil physicochemical properties but the associated changes in soil microbial communities and the mechanisms by which soil quality is improved by various plantations are not fully understood. In this study, soil physicochemical properties and microbial and enzyme activities were investigated in four forest types–*Castanopsis hystrix* pure forests (CHPF), *C. hystrix*–*Pinus elliottii* mixed forests (CHPEF), *C. hystrix*–*Michelia macclurei* mixed forests (CHMMF), and *C. hystrix*–*Mytilaria laosensis* mixed forests (CHMLF) in the subtropical region of China. The purpose of this study was to assess the effects of afforestation types on characteristics of soil–its physical, chemical, and biological properties. The results showed that the contents of soil total organic carbon (TOC), soil total nitrogen (TN), microbial biomass carbon (MBC), and microbial biomass nitrogen (MBN) were significantly improved in both CHMMF and CHMLF mixed forest stands when compared to the CHPF pure stand. Soil enzyme activities were enhanced in the mixed forests. In particular, high phosphatase activity was observed in CHMLF stands, leading to the transformation of soil phosphorus to available phosphorus in this forest type. Our study demonstrated that the broad–leaved mixed forests, but not coniferous mixed forests, could significantly improve soil quality in the study region. Our research provides a scientific insight into the promotion of vegetation restoration and plantation forest management in plantation regions of subtropical areas.

## 1. Introduction

Soils are the most important direct source of nutrients and water and play a key role in forest growth and biomass carbon accumulation [1,2]. Forest soil quality, especially the availability of soil nutrients and water, is an important factor affecting forest productivity [3]. Changes in soil quality can induce changes in ecological processes and the characteristics of forest ecosystems [4,5]. The changes of forest type can affect soil properties significantly [6]. Soil quality varies with different forest types, which alter the physical, chemical, biological, and biochemical properties of forest soils [7,8]. Soil enzyme activities are sensitive to the environment of different forest ecosystems and have the potential to serve as an indicator of the health and sustainability of managed ecosystems [9]. Having mixed tree species in plantations is a trend to promote carbon sequestration and enhance soil physicochemical properties, and microbial and enzyme activity is essential for soil biogeochemical processes in forest ecosystems [10].

Natural forest resources have declined considerably because of human activities and the growing demand for forest products. Globally, the forest area has decreased from 4.28 billion ha in 1990 to 3.99 billion ha in 2015 [11]. At the same time, the plantation forests have been expanded and reached over 278 million ha worldwide, about 78% of the global natural forest areas [11], providing about 46.3% of global industrial round wood products [12]. In China, the total plantation areas cover 69.3 million km^2^, which account for 36% of the national forest area and retain approximately 2.48 billion m^3^ of timber stocks [13]. Thus, plantations play an important role in forest resources, timber products, and ecosystem services on national and international scales [14,15]. A number of studies have shown that plantation type has a significant influence on soil’s structure, fertility, and quality as well ecological processes in forests [16,17]. When compared to the mono–species forests, planting mixed forests with multiple tree species can improve soil properties, such as soil pH [18,19,20,21], soil moisture, and bulk density [22], enhance soil microorganisms’ community and activity [23,24], accelerate the turnover of soil organic matter [25,26], and ameliorate soil nutrient status [27,28]. Zhou and Wang reported the C, N, and P contents of soil were greatly improved in *Picea koraiensis* mixed plantations compared to *P. koraiensis* mono–species forests [29]. Zhou et al. explored the difference in soil quality between pure and mixed Chinese fir forests, looking at the soil’s physical and chemical properties (*SBD*, SOM, TN, TP etc.), and found that these properties were better promoted in the mixed forests [30]. Soil microbiological and biochemical properties were significantly different between the mono–species pure forests and multiple–species mixed forests [31]. Singson et al. used soil microbiological and biochemical properties to compare the soil quality of different types of forest plantation in the northeastern Himalayas, India. The research shows that three different plantations (teak, arecanut, and rubber) significantly increased soil’s water holding capacity (WHC), soil organic carbon (SOC), and soil microbiological properties such as microbial biomass C and N (MBC and MBN) and percent of MBC to SOC [32]. However, the influence of planting various forest types on soil physical, chemical, and biological properties is still not fully understood. Soil enzymes are highly catalytically active substances produced by the action of soil microorganisms and are the main medium for controlling soil biochemical processes, such as nutrients (i.e., C, N, and P) cycling and SOM decomposition [33]. They play an irreplaceable role in maintaining soil physicochemical properties, fertility, and ecological health [34]. In recent years, soil enzyme activities have been considered an appropriate indicator of soil quality [35], as soil enzymes exhibit some correlation with soil changes and are closely related to soil quality. Therefore, in some studies, the physicochemical, biological, and biochemical properties of soils under the same site conditions of different plantations and natural forests, including nutrients, trace elements, microbial biomass, soil enzyme activity, etc., have been used to explore the soil quality of plantations [36].

In this study, soil properties and quality were examined in four forest types–*Castanopsis hystrix* pure forests (CHPF), *C. hystrix*–*Pinus elliottii* mixed forests (CHPEF), *C. hystrix*–*Michelia macclurei* mixed forests (CHMMF), and *C. hystrix*–*Mytilaria laosensis* mixed forests (CHMLF). The objectives of this project were (a) to investigate the differences in selected soil physicochemical properties and soil enzyme activities among four plantation types, (b) to explore the effects of four forest types on soil quality. We hypothesized that (1) there are significant differences in soil physicochemical properties and soil enzyme activities among the four plantation types, and (2) there are significant effects of four types of forest plantations on soil quality in the study area.

## 2. Results

### 2.1. Soil Physicochemical Properties Content

There were no significant differences between the soil pH values of the two layers except in the CHMLF stands(Table 1). The maximum soil pH value of the 0–20 cm layer occurred in CHPF and CHMLF stands and that of the 20–40 cm layer appeared in the CHPF pure stand. The soil SM was higher in the 0–20 cm layer than in the 20–40 cm layer in the same plantation type, but there were no significant differences between the two layers except in the CHMLF stands. The soil *SBD* content increased with the increase of soil layers in all stands. The highest soil *SBD* content of the 0–20 cm layer was in the CHPF pure stands and the lowest content of the 20–40 cm layer was in the CHMMF stands. The contents of TOC and TN both significantly decreased with increasing soil depth. The maximum TOC and TN contents of the two layers all existed in the CHMMF stands. The TP contents of the 0–20 cm layer were higher in the CHMMF stands than in the other stands but the only significant difference was found in the CHPEF stands. There were no significant differences of TP contents in the 20–40 cm layer in the four plantations.

### 2.2. Soil MBC and MBN Content

The quantitative relationships among MBC contents in the four plantations were similar in the 0–20 cm soil layer and the 20–40 cm soil layer (Figure 1a). The soil MBC content in CHMMF stands was higher but not significantly different to that in CHPEF stands. Soil MBC contents were significantly lower in CHPF and CHMLF stands than in the CHMMF stand. The highest MBN content of the 0–20 cm soil layer was in the CHMLF stands and that of the 20–40 cm layer was in the CHMMF stands (Figure 1b). However, there were no significant differences among the MBN contents of each soil layer in all plantations. No significant differences in MBC/MBN values were found in the 0–20 cm soil layer in all plantations. The MBC/MBN values of the 20–40 cm layer did not significantly differ among CHPF, CHPEF and CHMMF stands, but were significantly higher than those in the CHMLF stands.

### 2.3. Soil Enzyme Activities

The highest soil catalase activity was found in both 0–20 cm and 20–40 soil layers in CHMMF stands and there was no significant difference in soil catalase activity among the other examined stands (Figure 2a). The soil invertase activity of the 0–20 cm soil layer in CHMMF stands was higher than, but not significantly different to, that in the CHPF and CHPEF stands. The invertase activity was significantly lower in CHMLF stands than in CHMMF stands. The enzyme activities of the 20–40 cm soil layer decreased with the rank as CHPF > CHPEF > CHMMF > CHMLF, but no significant differences existed among these forest types (Figure 2b). The highest soil urease activity in both 0–20 cm and 20–40 soil layers was found in CHMMF stands, and there was no significant difference of soil urease activity among the other studied stands (Figure 2c). The soil phosphatase activities in the 0–20 cm and 20–40 cm layers were significantly higher in CHMMF and CHMLF than in the other two studied stands (Figure 2d). Four soil enzyme activities decreased with the increasing of soil layer in all examined forest types.

### 2.4. PCA of All Selected Soil Parameters in Two Soil Layers

The accumulative contributions of principal component 1 (PC1) and principal component 2 (PC2) to the selected soil properties were over 70% during the study, indicating that these two components could be more than 70% of the variables (Figure 3). Significant differences in spatial distribution were found among soil samples in the two soil layers in the four–plantation type. Soil samples in the 0–20 cm layer taken from CHPF stands were not affected by any specific factors (Figure 3a). However, soil samples were influenced primarily by invertase, MBC/MBN, and *SBD* in CHPEF, by *SM*, urease, TN, MBC, catalase, TP, and TOC in CHMMF, and by pH, phosphatase, and MBN in CHMLF stands, respectively. In addition, MBC, invertase, TP, MBC/MBN, pH, and urease had great effects on soil samples in the 20–40 cm layer of CHPF stands (Figure 3b). The key factor affecting soil samples in the 20–40 cm layer was only urease in CHPEF stands, TN, TOC, *SM*, MBC, and urease in CHMMF stands, and *SBD* and phosphatase in CHMLF stands.

### 2.5. Correlations among the Selected Soil Characters

There were positive relationships between the activities of catalase and urease (*p* < 0.05), catalase and phosphatase (*p* < 0.01), invertase and urease (*p* < 0.01), and urease and phosphatase (*p* < 0.01) in the studied forests (Table 2). The activities of catalase, invertase, urease, and phosphatase had positive relationships with the contents/values of MBN (*p* < 0.05) and TOC (*p* < 0.01) and TN (*p* < 0.01), MBC (*p* < 0.05), and MBN (*p* < 0.05) and TOC (*p* < 0.01) and TN (*p* < 0.01), MBC (*p* < 0.01) and MBN (*p* < 0.01) and TOC (*p* < 0.01) and TN (*p* < 0.01), MBN (*p* < 0.01) and TOC (*p* < 0.05), and TN (*p* < 0.01), respectively. The value of *SBD* had negative relationships with the contents of MBC, MBN, TOC, TN, and *SM* (*p* < 0.01), and negative relationships with catalase and phosphatase activities (*p* < 0.05). The value of pH had no relationship with other parameters. The content of *SM* had positive relationships with the contents of MBC, MBN, TOC, and TN (*p* < 0.01). There were positive relationships among MBC, MBN, TOC, and TN contents (*p* < 0.01). The contents of MBC, MBN, TOC, and TN had direct effects on soil enzyme activities, and the *SM* content and *SBD* values had indirect impacts on them by affecting MBC, MBN, TOC, and TN contents. The TP content was not a limiting factor in affecting soil enzyme activities.

## 3. Materials and Methods

### 3.1. Site Selection and Description

The studied area was located at the National Gaofeng Forest Farm in Nanning, Guangxi Zhuang Autonomous Region, China (22°50′–23°33′ N, 108°07′–109°21′ E). The landform is a low hilly and mountainous area with an elevation of 200–500 m. The annual average temperature is 21.35 °C and the annual rainfall is 1450 mm, with the rainfall from April to September accounting for more than 80%. The average annual evaporation is 1450 mm and the average annual relative humidity is 81%, with annual sunshine hours of 1600 h. The study area is a typical subtropical monsoon humid climate. The selected four plantation types in this study include *C. hystrix* pure forest (CHPF), *C. hystrix*–*Pinus elliottii* mixed forest (CHPEFF), *C. hystrix*–*Michelia macclurei* mixed forest (CHMMFF), and *C. hystrix*–*Mytilaria laosensis* mixed forest (CHMLFF). The study area was covered by evergreen broad–leaved natural forests with *C. hystrix* as the dominant species in the study sites. After clear–cutting, four types of forests were established with similar site conditions by using similar silviculture and forest management practices in March of 2007. Three 20 m × 20 m plots were set up as the replications for each of the plantation types so that a total of twelve sample plots were established in the study site. The diameter at breast height (DBH) and tree height of all trees were measured by circumference and clinometers in each plot of four plantation types (the site information was shown in Table 3). Understory shrubs were included–*C. hicklii, Ehretia thyrsiflora, Maesa parvifolia, Evodia lepta, Altingia chinensis,* and *Schefflera octophylla*. The herbivorous species at the study sites included *Sarcandra glabra, Pteris dissitifolia,* and *Mussaenda pubescens*.

### 3.2. Soil Sampling

Three soil profiles were randomly excavated in each plot and the soil samples were layered by a ring cutting machine according to depths of 0–20 cm and 20–40 cm, respectively [37]. The soil samples were brought back to the laboratory for further analysis.

### 3.3. Soil Physicochemical Properties Measurements

Soil samples were air dried and passed through a 100–mesh sieve. Soil pH was determined using a glass electrode method [38]. Soil bulk density (*SBD*) was determined using a common steel ring method [38]. The soil moisture content (*SM*) was determined using a drying method [38]. Soil total organic carbon (TOC) concentration was determined using the potassium dichromate oxidation–heating method [38]. The total nitrogen (TN) concentration in the soil was determined using the Kjeldahl semi–micro method [38]. The content of total phosphorus (TP) in the soil was determined using the molybdenum antimony colorimetric method [38].

### 3.4. Soil MBC and MBN Measurements

Microbial biomass carbon (MBC) and microbial biomass nitrogen (MBN) in the soil were determined using chloroform fumigation extraction. The fresh soil was sieved using a 2 mm sieve and four parts (10.000 g/part) were weighed repeatedly and placed in a vacuum dryer containing chloroform. The samples were cultured for 24 h in a constant temperature incubator at 25 °C and a chloroform–free fumigation control group was set up. Then, 40 mL of 0.5 mol/L K_2_SO_4_ solution was added to the soil samples and it was shaken at 300 rpm for 30 min, then it was filtered. The filtrate was determined using a total carbon–total nitrogen analyzer (Jena TOC Multi N/C 3100, Analytik Jena Instruments Ltd. Co., Jena, Germany) [38].

### 3.5. Soil Enzyme Activities Measurements

Fresh soil was used to measure soil enzyme activities. Four important potential enzymes activities of catalase, C–acquiring enzyme (invertase), N–acquiring enzyme(urease), and organic P–acquiring enzyme (phosphatase), were determined using the following modified methods: the potassium permanganate titration method, DNS colorimetry, sodium phenol colorimetry, and sodium phenylene phosphate colorimetry [39].

### 3.6. Statistical Analysis

All measurements of stand soils for each sample were averaged by three replicates and standard error (SE) was used when needed. Statistical tests for different stands were performed using one–way ANOVA. The statistical analysis was conducted using SPSS version 22.0. For multiple comparisons, treatment means were separated using least significant differences (LSD) at a *p* = 0.05 level. The soil physicochemical properties, MBC and MBN contents, and soil enzyme activities were subjected to principal component analysis (PCA) using CANOCO version 5.0 for each stand and to a Pearson correlation analysis.

## 4. Discussion

The influence of forest types, such as mono–species pure forests and multiple–species mixed forests, on soil nutrients is controversial [40,41,42]. The C and N contents were reduced in the topsoil in oak and Scots pine mixed forests compared to the pure forests [43], whereas red pine mixed forests can store more carbon in soils than pure forest stands in the USA [44]. No significant differences existed in soil C between mono–species plantations and multiple–species plantations in southeast China and southeast Queensland, Australia [45]. This controversial phenomenon may be caused by the different tree species composition of the studied forests and the original soil conditions in the study sites.

In this study, the soil TOC and TN contents were significantly higher in the *C. hystrix* and broad–leaved species mixed forests, such as CHMMF and CHMLF, than in CHPF and *C. hystrix* and coniferous species mixed forests, such as CHPEF. Litter production and decomposition might be the key factors regulating carbon turnover and nutrient recycling in forest ecosystems [46]. In general, the coniferous leaves have a coarse and hard texture, a high cellulose content, rich waxy cuticles, and poor water permeability, which may result in slow litter decomposition and affect the accumulation of organic matter in the soil [47]. Therefore, the litter decomposition rate might be one of the reasons for the differences in the soil nutrients of stands. In this study, the contents of soil organic carbon and total nitrogen were significantly negatively correlated with soil bulk density, indicating that the larger the soil bulk density, the more compact the soil, and the worse the soil ventilation, the more unfavorable to the decomposition and transformation of the litter. It resulted in a decrease in soil nutrients and a reduction of the energy of microbial self–synthesis and metabolism in the soil, and a decrease in the carbon and nitrogen content of soil microbial biomass [48]. A higher soil bulk density was found in CHPF and CHPEF stands in that research, which may be another reason.

Soil microbial biomass is the driving force of the transformation and cycling of soil organic matter and soil nutrients, which could have an essential effect on soil nutrient availability [49,50]. In our study, MBC and MBN contents were significantly positively correlated with soil TOC and TN. Furthermore, the results of PCA analysis indicated that TOC, TN, and MBC were major factors in CHMMF, but not in CHPF and CHPEF. Higher microbial biomass and soil moisture resulted in better soil nutrient states in CHMMF stands. MBN became the main factor in CHMLF stands. The difference in soil nutrient status between CHMMF and CHMLF may be caused by the composition and activity of the soil microbial community. Plants physiologically exuded a considerable number of organic compounds to the soil, which may modify microbial community composition and structure and promote microbial activity [51]. It is well known that different plants produce different kinds and quantities of secretions. In addition, microbial biomass was greatest in the topsoil in all studied forests. Topsoil has abundant fine root biomass, litter accumulation, and organic matter content, as well as good air–exchange conditions, which contribute to the growth and reproduction of soil microorganisms [52,53]. Along with the differences in soil nutrients and microbial biomass, the soil enzyme activities varied in the studied forests [54,55,56]. The activities of the four enzymes were almost the same in CHPF and CHPEF stands and were highest in the CHMMF stand. The results for enzyme activities were consistent with the findings for soil nutrient contents in this study. At the same time, the correlations between the soil enzymes involved in the nutrient cycle were consistent with the corresponding element content. Thus, the changes in enzyme activities were in agreement with the changes in soil nutrients. In addition, the high catalase enzyme activity indicated a better stability of soil microorganisms in CHMMF stands. In the CHMLF stand, the activities of soil phosphatase significantly increased even though the soil TP content was not altered. This phenomenon may be due to an increase in the proportion of soil–available phosphorus to total phosphorus, which is consistent with the results of other studies. For example, a study showed that soil TP contents were similar in Chinese fir and *Pinus* mixed forests and in Chinese fir pure forests, but available P contents were significantly higher in the former than in the latter forest type [45]. Some scholars pointed out that the introduction of broad–leaved trees increased tree diversity, improved soil microbial community composition and enzyme activity, as well as soil phosphorus availability [57].

## 5. Conclusions

In the present study, the *C. hystrix* and broad–leaved tree species mixed forests significantly improved soil physiochemical properties when compared to *C. hystrix* pure forests or *C. hystrix* and conifer mixed forests. Specifically, our study found that the lower the soil bulk density, the higher the soil TOC and TN contents, promoting the soil microbial and enzyme activities and thus improving the soil C and N contents in both CHMMF and CHMLF stands compared to CHPF pure stands. Soil quality, including soil C and N contents and soil microbial and enzyme activities, was not significantly altered in CHPEF when compared to CHPF. This phenomenon was likely related to the slow decomposition rate of tree litters from coniferous tree species. Our study demonstrated that the mixed forest with *C. hystrix and* broad–leaved tree species could greatly improve soil quality and fertility by increasing the soil microbial community and enzyme activities in the study region. The results provide a scientific basis for studying forest structure in terms of the species composition and ecosystem function of planted forests.

## Figures and Tables

**Figure 1 plants-12-02411-f001:**
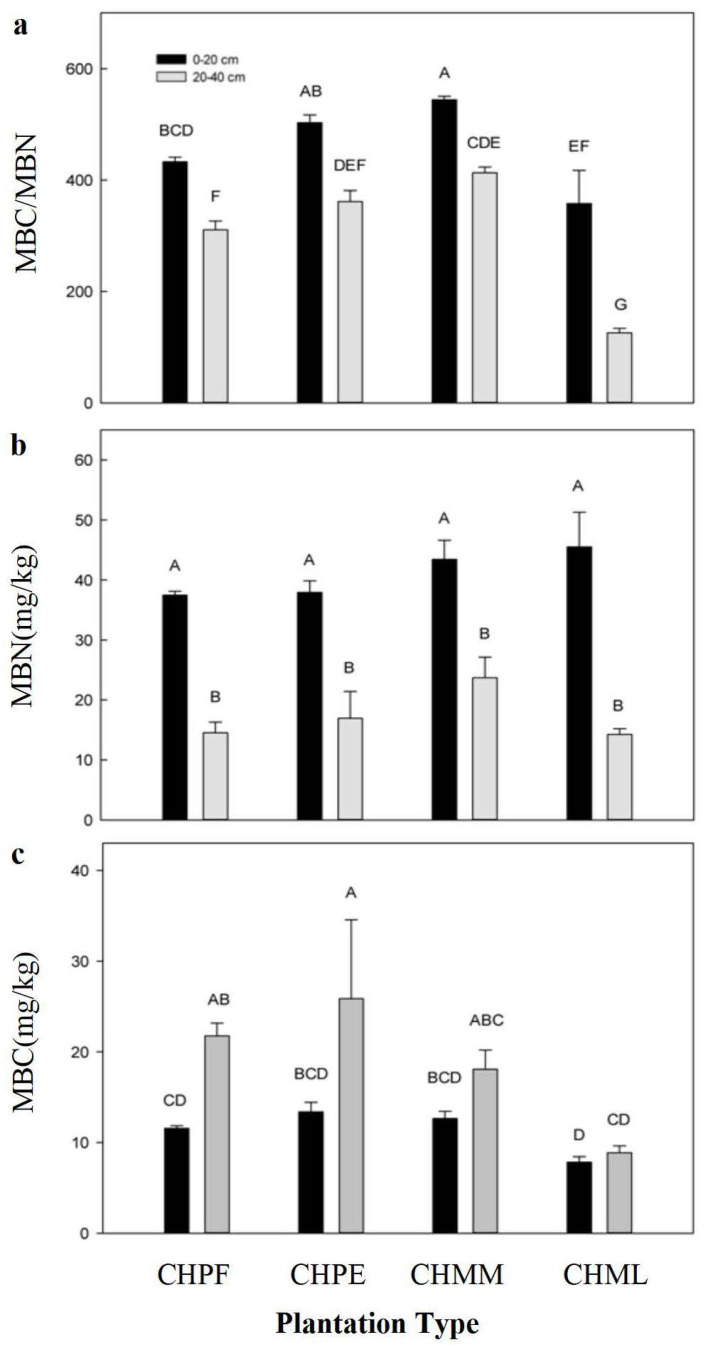
(**a**–**c**) Variation of MBC and MBN contents and MBC/MBN ratio in the four plantation types. Note: Uppercase letters (A–D) above the column indicate significant differences between the soil layers of different plantations (*p* < 0.05). *C. hystrix* pure forest (CHPF), *C. hystrix*–*Pinus elliottii* mixed forest (CHPEF), *C. hystrix*–*Michelia macclurei* mixed forest (CHMMF), and *C. hystrix*–*Mytilaria laosensis* mixed forest (CHMLF).

**Figure 2 plants-12-02411-f002:**
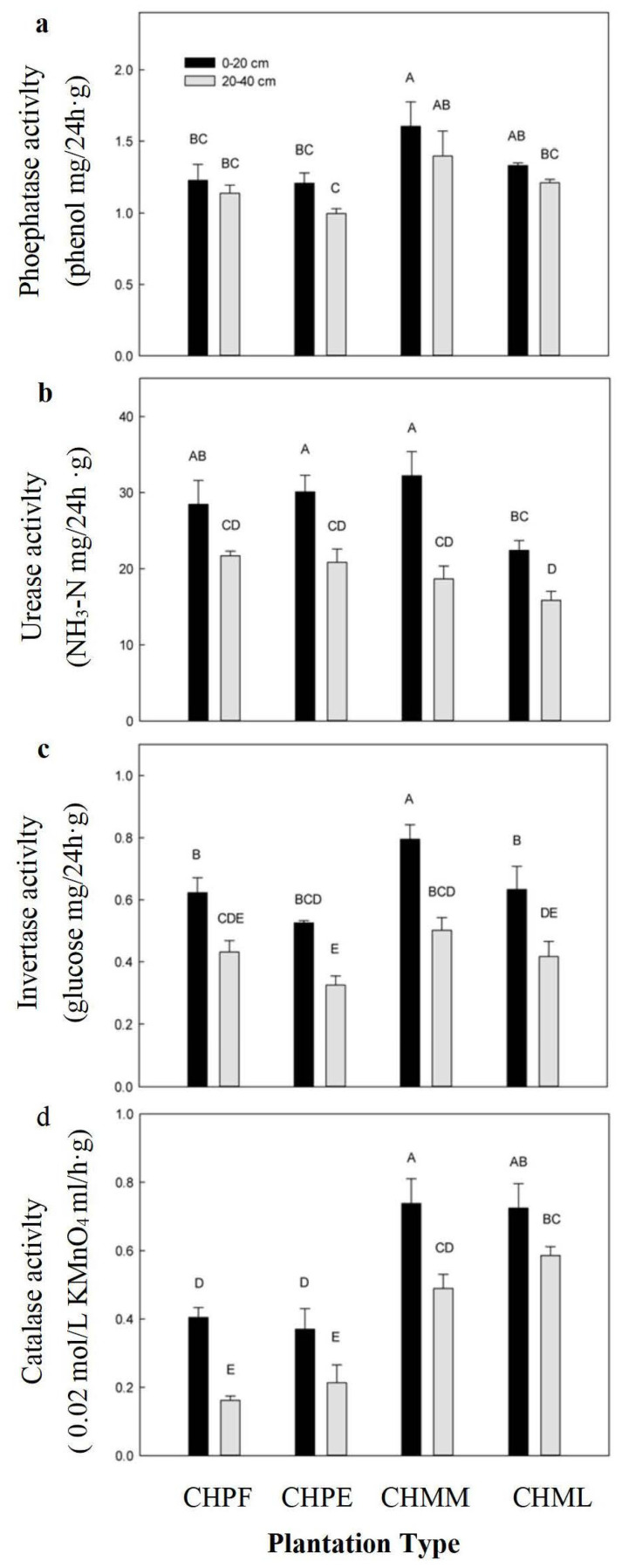
(**a**–**d**) Variation of soil enzyme activities in different soil layers in the four studied plantations. Note: Uppercase letters (A–D) above the column indicate significant differences between the soil layers of different plantations (*p* < 0.05). *C. hystrix* pure forest (CHPF), *C. hystrix*–*Pinus elliottii* mixed forest (CHPEF), *C. hystrix*–*Michelia macclurei* mixed forest (CHMMF), and *C. hystrix*–*Mytilaria laosensis* mixed forest (CHMLF).

**Figure 3 plants-12-02411-f003:**
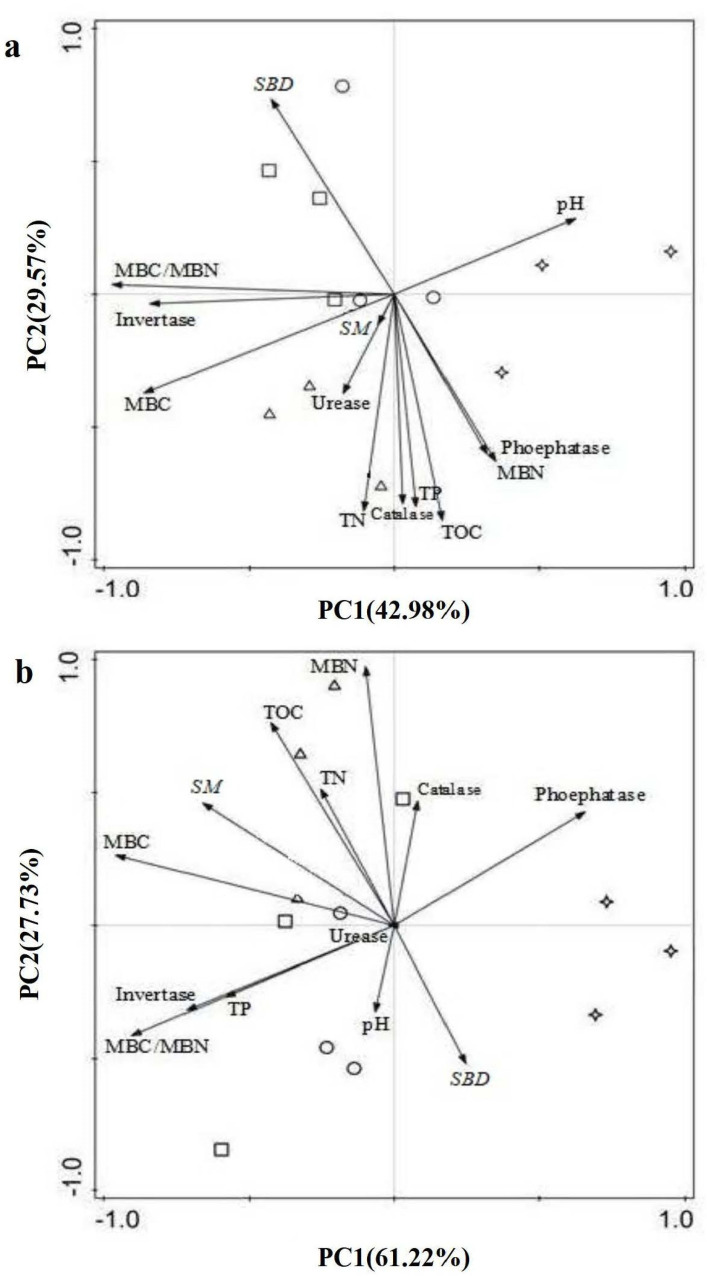
Principal component analysis (PCA) of selected soil biochemical and microbial properties, soil enzyme activities in 0–20 cm (**a**) and 20–40 cm (**b**) soil layers. Note: Different symbols indicate different plantation types. 
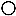
: CHPF, 
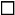
: CHPEF, 
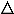
: CHMMF, and 
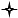
: CHMLF.

**Table 1 plants-12-02411-t001:** Soil physicochemical properties of different soil layers in the four studied plantations.

Plantation	Soil Layer (cm)	pH	*SM* (%)	*SBD* (g/cm^3^)	TOC (g/kg)	TN (g/kg)	TP (g/kg)
CHPF	0–20	4.32 ± 0.07A	19.90 ± 0.37AB	1.42 ± 0.04ABC	15.89 ± 2.71B	0.46 ± 0.02B	1.47 ± 0.31AB
	20–40	4.30 ± 0.01AB	17.95 ± 0.49BC	1.57 ± 0.02A	9.29 ± 1.53D	0.33 ± 0.04C	1.41 ± 0.47AB
CHPEF	0–20	4.11 ± 0.04C	20.86 ± 0.59A	1.33 ± 0.06CD	17.62 ± 1.51AB	0.47 ± 0.01B	0.97 ± 0.20B
	20–40	4.13 ± 0.10C	18.89 ± 0.72ABC	1.46 ± 0.05ABC	11.18 ± 0.70CD	0.33 ± 0.02C	1.26 ± 0.22AB
CHMMF	0–20	4.06 ± 0.03C	19.87 ± 1.06AB	1.25 ± 0.06D	21.01 ± 1.44A	0.66 ± 0.06A	1.81 ± 0.06A
	20–40	4.10 ± 0.01C	19.54 ± 0.51AB	1.38 ± 0.06BCD	15.20 ± 1.91BC	0.52 ± 0.06B	1.30 ± 0.11AB
CHMLF	0–20	4.32 ± 0.04A	20.22 ± 1.16A	1.23 ± 0.01D	18.95 ± 0.43AB	0.51 ± 0.02B	1.31 ± 0.04AB
	20–40	4.16 ± 0.04BC	16.79 ± 0.69C	1.53 ± 0.09AB	8.89 ± 0.67D	0.34 ± 0.04C	0.88 ± 0.11B

Note: Different uppercase letters (A–D) indicate significant differences between the same column at the *p* < 0.05 level. Soil moisture (*SM*), soil bulk density (*SBD*), total organic carbon (TOC), total nitrogen (TN), total phosphorus (TP).

**Table 2 plants-12-02411-t002:** Correlation matrix of the selected soil parameters in the studied forests.

Parameter	Catalase	Invertase	Urease	Phosphatase	MBC	MBN	pH	*SM*	*SBD*	TOC	TN	TP
Invertase	0.153	1										
Urease	0.509 *	0.683 **	1									
Phosphatase	0.552 **	0.143	0.609 **	1								
MBC	0.357	0.732 **	0.520 **	0.117	1							
MBN	0.461 *	0.583 **	0.654 **	0.573 **	0.673 **	1						
pH	−0.091	0.022	−0.004	−0.198	−0.247	0.034	1					
*SM*	0.312	0.404	0.277	0.161	0.745 **	0.730 **	−0.015	1				
*SBD*	−0.559 **	−0.331	−0.454 *	−0.497 *	−0.565 **	−0.775 **	0.065	−0.658 **	1			
TOC	0.640 **	0.490 *	0.572 **	0.500 *	0.715 **	0.860 **	−0.090	0.662 **	−0.860 **	1		
TN	0.793 **	0.406 *	0.681 **	0.555 **	0.644 **	0.683 **	−0.168	0.557 **	−0.803 **	0.826 **	1	
TP	0.368	0.324	0.340	0.033	0.336	0.189	0.246	0.118	−0.239	0.375	0.432 *	1
MBC/MBN	−0.305	−0.101	−0.352	−0.623 **	0.083	−0.573 **	−0.170	−0.154	0.332	−0.389	−0.285	0.184

Note: * indicates significant correlation at *p* < 0.05 level, ** indicates significant correlation at *p* < 0.01 level.

**Table 3 plants-12-02411-t003:** Characteristics of the four plantations in study sites.

Forest Type	Main Tree Species	Canopy Density (%)	Mean Height (m)	Mean DBH (cm)
CHPF	*C. hystrix*	85	14.4 ± 0.2A	17.2 ± 0.3A
CHPEF	*C. hystrix* +	85	15.6 ± 0.4A	18.6 ± 0.5B
	*Pinus elliottii*		15.2 ± 0.3A	21.1 ± 0.2C
CHMMF	*C. hystrix* +	90	15.6 ± 0.1A	14.7 ± 0.1D
	*Michelia macclurei*		14.3 ± 0.1A	14.8 ± 0.2D
CHMLF	*C. hystrix* +	90	15.2 ± 0.2A	12.4 ± 0.2E
	*Mytilaria laosensis*		17.9 ± 0.1B	15.6 ± 0.7D

Different uppercase letters (A–D) indicate significant differences between the same column at the *p* < 0.05 level.

## Data Availability

The data presented in this article are available on request from the corresponding authors.

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
