# Peer review of "Changes in Soil Properties, Microbial Quantity and Enzyme Activities in Four Castanopsis hystrix Forest Types in Subtropical China"

_plants, 2023, doi:10.3390/plants12132411_

Round 1
Reviewer 1 Report
This is a simple and straightforward manuscript that has worthy implications on a fairly small scale. The results are not surprising but the application to this particular set of forest types is interesting and illuminating. The enzymes chosen do not represent as broad a range of microbial enzymes as would be helpful to make the general conclusions that are stated. The manuscript contains some awkward English language constructions and I will highlight a few of them though I do not profess to be a professional copy editor. I will go through these line by line in the first part of the paper to illustrate my concerns with no claim to thoroughness.
l 17-18 property and activity should both be plural.
l 25 should read "enzyme activities were"
l 33 should read "have declined considerably"
l 38 million hectares?
l 41 international scales
l 60 among four plantation types
l 61 four plantation types ?
Section 2
Need more information on prior land use history, planting dates of current plantations to determine plot ages, spacing of plantations, productivity measures, other silvicultural data and Table 1 would be more useful with biomass or volume estimates and age of plantations
l 70-71 change among which the to "with" and accounts to "accounting'
l 74 CHPF should be CHPE I believe
l 80 foure should be "four"
l 82 Pubescens should be in lower case
Section 2.2 Was there no litter layer? Why were the particular depths selected? Were they appropriate across all plots and how do they relate to observed soil horizon
l 109 awkward "The sample was added...." is not clear similarly in line 110 "the filtrate was determined is not proper construction
Section 2.4 Somewhere in the manuscript it would be helpful to elaborate on why these particular enzymes were selected for study and other were not included For example why were cellulases not included
Figure 1. Should indicate the layers in this figure as you did in Figure 2.
Figure 2 The x axis is not aligned properly with the figure
Figure 3 Consider dashed lines around each cluster of plantations types to clarify results of the PCA
Tables 2 and 3 Consider replacing theta g and rho b with SM (soil moisture) and BD (bulk density)
l 267 "enrich fine roots biomass" unclear
l 285 as well as ..add "as"
l 290 he should be the
Section 5 What is the evidence for increase fertility or even productivity
see above
Author Response
I would like to thank you for your invaluable comments and suggestions on our manuscript.
Based on your invaluable comments and suggestions, we have carefully checked throughout the manuscript and revised the manuscript.
Please see the attachment for the point-by-point response.

Reviewer 2 Report
Growing demand for forest products and carbon sequestration in forest ecosystems while avoiding over-exploitation of natural forests, planted forests are growing rapidly as a major component of global afforestation/reforestation However, there are serious concerns about mono-species plantations, including declines in stand productivity and concentration nutrients in the soil. The advantages of mixed plantations over monocultures have been widely recognized, but the mechanisms behind these differences are poorly understood.
The manuscript submitted for review is interesting from the scientific and application point of view.
The paper is well structured, the discussions are in comparison with other researches and results. The references are adequate to the subject. I find the conclusions clear and concise.
Also, very minor corrections are needed:
1. In the summary of sentences 2 “In this study…..” and 2 “The purpose of this study…. require rewording because they are a significant repetition of information.
2. Page 2, lines 60 and 61 “b) to assess the soil quality of the four plantations, and c) to explore the effect of four plantations on soil quality” - in my opinion point c is enough
3. Page 3 “2.3. Soil Physicochemical Properties Measurements” - describe the methods in detail or provide reference literature
Author Response

(The authors gave the same response as above.)

Reviewer 3 Report
Dear Authors,
Previous studies have shown that plants often have species-specific effects on soil properties. There is still a lack of information on the effects of various tree species on maximising soil function (e.g. enhancing carbon (C) and nitrogen (N) storage, promoting nutrient cycling and water storage), and especially on the changes in soil microbial communities. Forests affect the composition of microbial communities not only directly, but also indirectly through changes in soil chemical and physical properties depending on the forest type, biodiversity, and land use history. Generally, the composition of microbial communities formed under broadleaf forests is radically different from those formed under coniferous species.
In a manuscript submitted by Renjie Wang et al. the impact of afforestation types on the physical, chemical and biological properties of soils was assessed.
In the literature on the subject, the authors addressed this problem. In this respect, the work is not original.
In this review, I offer a few suggestions as to where certain points can be elaborated upon or revised in the manuscript.
Comments and suggestions:
1. Line 79: Please add information. What device was used to measure the height?
2. Line 91: Why were soil samples taken at 0-20 and 20-40 cm? Add a literature reference or justify such a methodology?
3. Line 111: What country is this device manufactured in?
4. Lines 200-201- Fig.3. Adjust the font size which is disproportionate.
5. Lines 288-298: Conclusions: Specifically how and where these results can be used. Furthermore, please consider that the conclusion is intended to help the reader understand why your research should matter to them. A conclusion is not merely a summary of the main results but a synthesis of key points and where you recommend new areas for future research.
6. Reference Lines 313-418: References not prepared in accordance with the requirements journal Plants. Plants | Instructions for Authors (mdpi.com)
Author 1, A.B.; Author 2, C.D. Title of the article. Abbreviated Journal Name Year, Volume, page range.

Author Response

(The authors gave the same response as above.)

Reviewer 4 Report
The work is interesting, although the formulation of hypotheses would add value to it. Although the research results are not unambiguous, they provide additional information on the functioning of the soil and the organisms directly responsible for the decomposition of organic matter - bacteria. The results of the study add to our understanding of soil health in forest plantings in the subtropical zone.
Line 58 and line 62: The aims of the study a) and c) sound very similar. It is worth considering formulating one goal. It would be best to formulate hypotheses to emphasize the importance of the research.
Line 76: …‘These forests are planned in the same site condition in 2007 with using similar silviculture and management practice.’ Probably …‘These forests are planted in the same site condition in 2007 with using similar silviculture and management practice.’
Line 84: Table 1- please give information about unit of the canopy density
Line 290: …..Specifically, he lower…. Should be …..Specifically, the lower….

Author Response

(The authors gave the same response as above.)
